# Anomalous normal fluid response in a chiral superconductor UTe$_2$

Seokjin Bae [1,4✉], Hyunsoo Kim [1,5], Yun Suk Eo[1], Sheng Ran [1,2,6], I-lin Liu [1,2], Wesley T. Fuhrman[1], Johnpierre Paglione [1,3], Nicholas P. Butch[1,2] & Steven M. Anlage [1✉]

Chiral superconductors have been proposed as one pathway to realize Majorana normal fluid at its boundary. However, the long-sought 2D and 3D chiral superconductors with edge and surface Majorana normal fluid are yet to be conclusively found. Here, we report evidence for a chiral spin-triplet pairing state of UTe$_2$ with surface normal fluid response. The microwave surface impedance of the UTe$_2$ crystal was measured and converted to complex conductivity, which is sensitive to both normal and superfluid responses. The anomalous residual normal fluid conductivity supports the presence of a significant normal fluid response. The superfluid conductivity follows the temperature behavior predicted for an axial spin-triplet state, which is further narrowed down to a chiral spin-triplet state with evidence of broken time-reversal symmetry. Further analysis excludes trivial origins for the observed normal fluid response. Our findings suggest that UTe$_2$ can be a new platform to study exotic topological excitations in higher dimension.

[1] Maryland Quantum Materials Center, Department of Physics, University of Maryland, College Park, MD, USA. [2] NIST Center for Neutron Research, National Institute of Standards and Technology, Gaithersburg, MD, USA. [3] The Canadian Institute for Advanced Research, Toronto, ON, Canada. [4] Present address: Materials Research Laboratory, University of Illinois Urbana-Champaign, Urbana, IL, USA. [5] Present address: Department of Physics and Astronomy, Texas Tech University, Lubbock, TX, USA. [6] Present address: Department of Physics, Washington University, St. Louis, MO, USA. ✉email: sjbae@terpmail.umd.edu; anlage@umd.edu

Topological insulators, with nonzero topological invariants, possess metallic states at their boundary[1]. Chiral superconductors, a type of topological superconductors with nonzero topological invariants, possess Majorana fermions at their boundary[2–5]. Majorana fermions are an essential ingredient to establish topological quantum computation[6]. Hence, great effort has been given to search for chiral superconducting systems. So far, evidence for the 1D example has been found from a semiconductor nanowire with end-point Majorana states[7]. However, 2D and 3D chiral superconductors with a surface Majorana normal fluid has not been unequivocally found[3]. Recently, a newly discovered heavy-fermion superconductor UTe$_2$ (ref. [8]) is proposed to be a long-sought 3D chiral superconductor with evidence of the chiral in-gap state from a scanning tunneling microscopy (STM) study[9]. This raises a great deal of interest in the physics community to independently establish the existence of the normal fluid response, determining whether or not the response is intrinsic, and identifying the nature of the pairing state of UTe$_2$.

To address these three questions, the microwave surface impedance of a UTe$_2$ crystal was measured by the dielectric resonator technique (see Supplementary Notes 1–3). The obtained impedance was converted to the complex conductivity, where the real part is sensitive to the normal fluid response and the imaginary part is sensitive to the superfluid response. By examining these results, here we confirm the existence of the significant normal fluid response of UTe$_2$, verify that the response is intrinsic, and identify that the gap structure is consistent with the chiral spin-triplet pairing state.

## Results

**Anomalous residual normal fluid response.** Figure 1 shows the surface impedance $Z_s = R_s + iX_s$ of the sample as a function of temperature, measured from the disk dielectric resonator setup (Supplementary Note 2 and ref. [10]). The surface resistance $R_s$ decreases monotonically below ≈1.55 K and reaches a surprisingly high residual value $R_s(0) ≈ 14$ mΩ at 11.26 GHz. This value is larger by an order of magnitude than that of another heavy-fermion superconductor CeCoIn$_5$ ($R_s(0) ≈ 0.9$ mΩ at 12.28 GHz)[11]. Subsequently its electrical resistance was determined by transport measurement and a midpoint $T_c$ of 1.57 K was found.

With the surface impedance, the complex conductivity $\tilde{\sigma} = \sigma_1 - i\sigma_2$ of the sample can be calculated. In the local electrodynamics regime (Supplementary Note 4), one has

$Z_s = \sqrt{i\mu_0\omega/\tilde{\sigma}}$. Figure 2a shows $\sigma_1$ and $\sigma_2$ of the sample as a function of temperature. Here, an anomalous feature is the monotonic increase of $\sigma_1(T)$ as $T$ decreases below $T_c$. Note that $\sigma_1$ is a property solely of the normal fluid. For superconductors with a topologically trivial pairing state, most of the normal fluid turns into superfluid and is depleted as $T \to 0$. As a result, in the low-temperature regime, $\sigma_1$ shows a strong decrease as temperature decreases, and is expected to reach a theoretically predicted residual value $\sigma_1(0)/\sigma_1(T_c) = 0$ (for fully gapped $s$-wave[12]), <0.1 (for the bulk state of a point nodal $p$-wave[13]), and 0.1–0.3 (for line nodal $d_{x^2-y^2}$-wave[14,15]). As shown in Fig. 2b, this behavior is observed for the case of Ti[16] ($s$-wave), as well as CeCoIn$_5$ (ref. [11], $d_{x^2-y^2}$-wave). In contrast, the UTe$_2$ crystal shows a monotonic increase in $\sigma_1$ as the temperature decreases and reaches a much larger $\sigma_1(0)/\sigma_1(T_c) = 2.3$, implying the normal fluid conduction channel is still active and provides a significant contribution even at the lowest temperature.

**Axial triplet pairing state from superfluid response.** Another property one can extract from the complex conductivity is the effective penetration depth. The imaginary part of the complex conductivity $\sigma_2(T)$ determines the absolute value of the effective penetration depth at each temperature as $\sigma_2(T) = 1/\mu_0\omega\lambda_{eff}^2(T)$. Once the absolute value of the penetration depth is known, the normalized superfluid density can be calculated as $\rho_s(T) = \lambda_{eff}^2(0)/\lambda_{eff}^2(T)$ (see "Methods" for how $\lambda_{eff}(0)$ is determined), and its low-temperature behavior is determined by the low-energy excitations of the superconductor, which is sensitive to the pairing state[17]. The $s$ and $d$-wave pairing states, representative spin-singlet pairing states, are inconsistent with our penetration depth data (see Supplementary Note 5). More crucially, singlet states cannot explain either the reported upper critical field $H_{c2}$, which is larger than the paramagnetic limiting field[8], or the absence of a change in the Knight shift across the $T_c$ (refs. [8,18]). Thus, only the spin-triplet pairing states are discussed below.

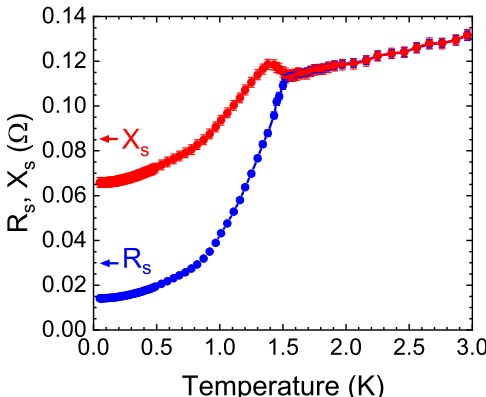

**Fig. 1 Microwave surface impedance of a UTe$_2$ single crystal.** The measured temperature dependence of the surface impedance of a UTe$_2$ sample at 11.26 GHz. The blue curve represents the surface resistance $R_s$ and the red curve represents the surface reactance $X_s$. The determination of the error bars of $R_s$ and $X_s$ is described in the Supplementary Note 1.

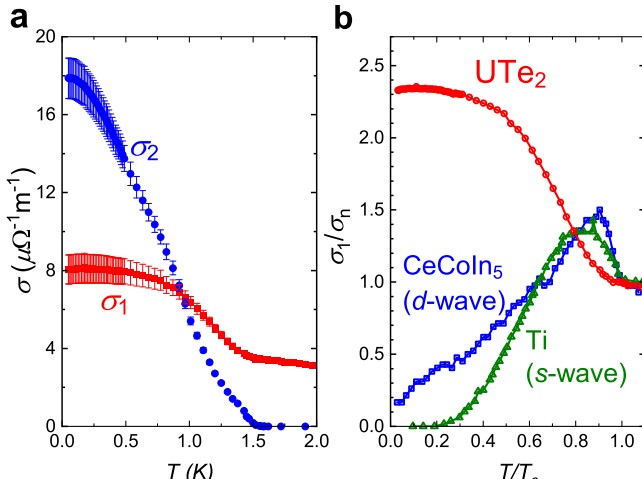

**Fig. 2 Anomalous residual conductivity of UTe$_2$ compared to superconductors with other pairing states. a** Real (red) and imaginary (blue) part of the complex conductivity $\tilde{\sigma} = \sigma_1 - i\sigma_2$ of the UTe$_2$ sample at 11.26 GHz. **b** Normalized (by $\sigma_n = \sigma_1(T_c)$) real part of conductivity of UTe$_2$ (red), a line nodal $d_{x^2-y^2}$-wave superconductor CeCoIn$_5$ (blue)[11], and a fully gapped $s$-wave superconductor Ti (green)[16] versus reduced temperature $T/T_c$. All measurements are done with the same, low frequency-to-gap ratio of $\hbar\omega/2\Delta_0 \approx 0.08$. Note that the error bar of the $\tilde{\sigma}$ is propagated from that of the $Z_s$ in Fig. 1.

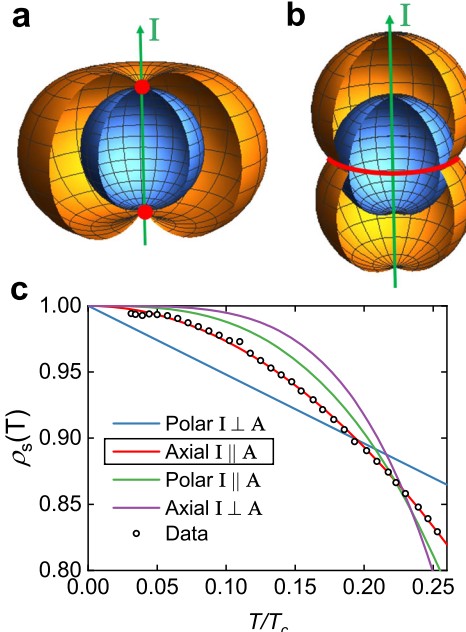

**Fig. 3 Evidence for the axial triplet pairing state from the superfluid density. a** A schematic plot of the gap magnitude $|\Delta(\mathbf{k})|$ (orange) and the Fermi surface (blue) in momentum space for the axial triplet pairing state. $\mathbf{I}$ represents the symmetry axis of the gap function. Note that two point nodes (red) exist along the symmetry axis. **b** For the case of the polar state. A line node (red) exists along the equatorial plane. **c** Low-temperature behavior of the normalized superfluid density $\rho_s(T)$ in UTe$_2$ with best fits for various triplet pairing states, and relative direction between the symmetry axis $\mathbf{I}$ and the vector potential $\mathbf{A}$. Since $\mathbf{I}$ connects the two point nodes of the gap of the axial pairing state and the measurement surveys the ab-plane electrodynamics, one can conclude that the point nodes are located near the ab-plane. Evidence of broken time-reversal symmetry[20] further narrows down the pairing state from axial to chiral.

For a spin-triplet pairing state, $\rho_s(T)$ follows different theoretical low-temperature behaviors depending on two factors[17,19]. One is whether the magnitude of the energy gap $|\Delta(\hat{\mathbf{k}}, T)|$ follows that of an axial state $\Delta_0(T)|\hat{\mathbf{k}} \times \hat{\mathbf{I}}|$ (Fig. 3a) or a polar state $\Delta_0(T)|\hat{\mathbf{k}} \cdot \hat{\mathbf{I}}|$ (Fig. 3b), where $\Delta_0(T)$ is the gap maximum. The other is whether the vector potential direction $\hat{\mathbf{A}}$ is parallel or perpendicular to the symmetry axis of the gap $\hat{\mathbf{I}}$. Figure 3c shows fits of $\rho_s(T)$ to the theoretical behavior (refs. [17,19] and Supplementary Note 8) of the various triplet pairing states. Apparently, the data follows the behavior of the axial pairing state with the direction of the current aligned to $\hat{\mathbf{I}}$. The axial state can be either chiral or helical depending on the presence or absence of time-reversal symmetry breaking (TRSB) of the system[3]. Recently, direct evidence for TRSB in this system was found by a finite polar Kerr rotation angle developing below $T_c$ (ref. [20]). Also, a specific heat study, a sensitive probe to resolve multiple superconducting transitions, showed two jumps near 1.6 K (ref. [20]). This implies that two nearly degenerate order parameters coexist, allowing the chiral pairing state from the group theory perspective[20]. Therefore, one can argue that UTe$_2$ shows $\rho_s(T)$ consistent with the chiral triplet pairing state. In addition, since the symmetry axis connects the two point nodes of the chiral pairing order parameter, and the measurement surveys the ab-plane electrodynamics, one can further conclude that the point nodes are located near the ab-plane. The low-temperature asymptote of

$\rho_s(T)$ in this case is given as $\rho_s(T) = 1 - \pi^2(k_B T / \Delta_0(0))^2$. The fitting in Fig. 3c yields an estimate for the gap size $\Delta_0(0) = 1.923 \pm 0.002 k_B T_c \approx 0.265$ meV. Note that a recent STM study[9] measures a similar gap size (0.25 meV).

**Examination of extrinsic origins.** Our study shows evidence for the chiral triplet pairing state and a substantial amount of normal fluid response in the ground state of UTe$_2$. Before attributing this residual normal fluid response to an intrinsic origin, one must first examine the possibilities of an extrinsic origin. One of the possible extrinsic origins would be a large bulk impurity scattering rate $\Gamma_{imp}$. However, if one fits the temperature dependence of the normalized superfluid density to $\rho_s(T) = 1 - aT^2$ and compares the estimated coefficient $a$ with the modified theoretical asymptote for the chiral triplet state, which includes $\Gamma_{imp}$ (ref. [19]),

$$\rho_s(T) = 1 - \frac{1}{1 - 3\frac{\Gamma_{imp}}{\Delta_0(0)}\left(\frac{\pi}{2}\ln 2 - 1\right)} \frac{\pi^2}{1 - \frac{\pi\Gamma_{imp}}{2\Delta_0(0)}}\left(\frac{k_B T}{\Delta_0(0)}\right)^2, \quad (1)$$

one obtains a quadratic equation for $(\Gamma_{imp}, \Delta_0(0))$, and two solutions: $\Gamma_{imp} \leq 0.06\Delta_0(0)$ and $\Gamma_{imp} \geq 4.36\Delta_0(0)$ for the range of $\Delta_0(0) \leq 0.280$ meV. For a nodal superconductor, the large $\Gamma_{imp} \geq 4.36\Delta_0(0)$ suppresses $T_c$ more than 80% (refs. [21,22]). Such suppression was not observed for the samples grown by the chemical vapor transport method[8,20]. They show a consistent $T_c \sim 1.6$ K, including the sample in this study. In contrast, the samples prepared by the flux-growth method[23] show suppression in $T_c$ all the way down to <0.1 K, as their normal state dc resistivity is much higher. This observation leaves the small $\Gamma_{imp} < 0.06\Delta_0$ solution as the only physically reasonable choice in our sample. Note that this small bulk $\Gamma_{imp}$ is consistent with the absence of the residual linear term in the thermal conductivity[24], both implying the clean limit $\Gamma_{imp} \ll 2\Delta_0(0)$. These results are inconsistent with the impurity-induced bulk normal fluid scenario. Note that the temperature independent $\Gamma_{imp}$ here is much different from the scattering rate above $T_c$ (see Supplementary Note 4), possibly suggesting the presence and dominance of highly temperature-dependent inelastic scattering due to spin fluctuations in the normal state.

Another possibility is a quasiparticle response excited by the microwave photons of the measurement signal. However, this scenario is also improbable because the maximum energy gap $\Delta_0(0) = 265$ μeV is much larger than that of the microwave photon $E_{ph} = 45$ μeV used here. At this low $E_{ph}/\Delta_0(0)$ ratio, even when the upper limit of $\Gamma_{imp} = 0.06\Delta_0(0) \sim 0.1 k_B T_c$ is assumped, a theoretical estimate[13] predicts only a small residual normal response from the bulk states $\sigma_1(0)/\sigma_n < 0.1$, which cannot explain the measured value of $\sigma_1(0)/\sigma_n = 2.3$.

**Possible intrinsic origin.** With several candidates for extrinsic origin excluded, now one can consider the possibility of an intrinsic origin. One important constraint to consider is the recent bulk thermal conductivity measurement in UTe$_2$ (ref. [24]). It revealed the absence of a residual linear term as a function of temperature in the thermal conductivity, implying the absence of residual normal carriers, at least in the bulk (see Supplementary Note 7). This result suggests the microwave conductivity data can be explained by a combination of a surface normal fluid and bulk superfluid. Topologically trivial origins for the surface normal fluid (e.g., pair breaking rough surface) are first examined, and shown to be inconsistent with the monotonic increase of $\sigma_1$ as $T \to 0$ in Fig. 2b (see Supplementary Note 11). Instead, considering the evidence of the chiral triplet pairing state from the superfluid density analysis and polar Kerr rotation measurement[20], a more pluausible source of

the surface normal fluid is the gapless chiral-dispersing surface states of a 3D chiral superconductor[4]. Point nodes of a chiral superconductor can possess nonzero Chern number with opposite sign. These point nodes in the superconducting gap are analogous to the Weyl points in the bulk energy bands of a Weyl semimetal. The nodes are predicted to introduce gapless surface Majorana arc states that connect them[5]. Evidence for these surface states in UTe$_2$ is seen in a chiral in-gap density of states from an STM study[9].

If this scenario is true, the anomalous monotonic increase in $\sigma_1$ down to zero temperature from this surface impedance study (Fig. 2b) may be understood as the enhancement of the scattering lifetime of the surface normal fluid. With its chiral energy dispersion, direct backscattering can be suppressed. As the temperature decreases, the superconducting gap $\Delta_0(T)$ that provides this topological protection increases, while thermal fluctuations $k_B T$ which poison the protection decrease. As a result, the suppression of the backscattering becomes stronger as $T \to 0$, which can end up enhancing the surface scattering lifetime (and $\sigma_1$). Although this argument is speculative at the moment, we hope the anomalous behavior of the $\sigma_1(T)$ reported in this work motivates quantitative theoretical investigation for the microwave response of the topological surface state of chiral superconductors.

In conclusion, our findings imply that UTe$_2$ may be the first example of a 3D chiral spin-triplet superconductor with a surface Majorana normal fluid. With topological excitations in a higher dimension, this material can be a new platform to pursue unconventional superconducting physics, and act as a setting for topological quantum computation.

## Methods

**Growth and preparation of UTe$_2$ single crystals.** The single-crystal sample of UTe$_2$ was grown by the chemical vapor transport method, using iodine as the transport agent[8]. For the microwave surface impedance measurement, the top and bottom $ab$-plane facets were polished on a 0.5 μm alumina polishing paper inside a nitrogen-filled glove bag (O$_2$ content < 0.04%). After polishing was done, the sample was encapsulated by Apeizon N-grease (see Supplementary Note 10) before being taken out from the bag, and then mounted to the resonator so that the sample is protected from oxidization. Long-term storage of the sample is done in a glove box with O$_2$ content < 0.5 p.p.m. Note that the electrical properties of oxidized uranium are summarized in Supplementary Note 9. The sample size after polishing is about ~1.5 × 0.7 × 0.3 mm$^3$ with the shortest dimension being along the crystallographic $c$-axis of the orthorhombic structure. The midpoint $T_c$ of the sample from DC transport measurements was 1.57 K.

**Microwave surface impedance measurement.** Due to its large volume, the measurement setup, data processing procedure, and interpretation are decribed in detail in the Supplementary Notes 1 and 2.

**Determining of the value of the zero temperature absolute penetration depth and comparison to other uranium-based superconductors.** The effective penetration depth at each temperature ($T \geq 50$ mK) can be obtained from $\sigma_2(T) = 1/\mu_0\omega\lambda_{\text{eff}}^2(T)$, where $\sigma_2(T)$ is obtained by the surface impedance $Z_s(T)$ data. The effective penetration depth at zero temperature can be obtained by extrapolating the data with a power law fit $\lambda_{\text{eff}}(T) - \lambda_{\text{eff}}(0) = aT^c$ over the low-temperature regime $T < 0.3T_c$, resulting in $\lambda_{\text{eff}}(0) = 791$ nm and $c = 2.11$. This value is similar to those found in the uranium-based ferromagnetic superconductor series, such as UCoGe ($\lambda_{\text{eff}}(0) \sim 1200$ nm)[25] and URhGe ($\lambda_{\text{eff}}(0) \sim 900$ nm)[26], where UTe$_2$ represents the paramagnetic end member of the series[8]. This result is also consistent with recent muon-spin rotation measurements on UTe$_2$, which concluded $\lambda_{\text{eff}}(0) \sim 1000$ nm (ref. [27]).

**Error bar of the fitting parameters of the normalized superfluid density.** In the fitting of the normalized superfluid density, the error bar of the fitting parameter (e.g., $\Delta_0$) was determined by the deviation from the estimated value, which increases the root-mean-square error of the fit by 1%.

## Data availability

The datasets generated and analysed in this work are provided with the paper in the Source data tab. Source data are provided with this paper.

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

## Acknowledgements

This work is supported by NSF grant No. DMR-1410712 (Support of S.B.), DMR-2004386 (data analysis), DOE grants No. DE-SC 0017931 (support of S.A.), DE-SC

0018788 (support of S.B. for low-temperature surface impedance measurements), DE-SC-0019154 (low-temperature transport measurements), Gordon and Betty Moore Foundation's EPiQS Initiative through Grant GBMF9071 (single-crystal synthesis), NIST (crystal synthesis and sample characterization), and the Maryland Quantum Materials Center (facilities support). Identification of commercial materials does not imply endorsement by NIST.

## Author contributions

S.B., N.P.B., and S.M.A. conceived the project. S.B. polished the sample, conducted the microwave surface impedance measurements, and carried out analysis on the complex conductivity. H.K. and Y.S.E. performed the transport measurements. S.R., I.L., and W.T.F. grew the UTe$_2$ single crystal used in this study. S.B., H.K., J.P., N.P.B., and S.M.A. interpreted the results. S.B. and S.M.A. wrote the manuscript with input from other authors.

## Competing interests

The authors declare no competing interests.
