## [Peer Review File · Nature Communications]

Editorial Note: This manuscript has been previously reviewed at another journal that is not operating a transparent peer review scheme. This document only contains reviewer comments and rebuttal letters for versions considered at Nature Communications. Mentions of the other journal have been redacted. Parts of this peer review file have been redacted as indicated to remove third-party material where no permission to publish could be obtained.

REVIEWER COMMENTS

Reviewer #1 (Remarks to the Author):

I have carefully read the revised version of this manuscript, which was previously submitted to [Redacted]. In their Reply the authors have taken away most of the concerns raised in my report. They have also made a number of clarifications in the extensive SM file. Overall, the authors have improved the manuscript. I still find it difficult to digest that other possible causes for the normal contribution are ruled out without much certainty. Especially, the authors rely on Metz et al. to explain the low-temperature contribution in the specific heat as being not relevant in the current experiment. Nonetheless, UTe₂ is certainly an unconventional superconductor. The current paper helps to unravel its nodal gap structure and is of interest to the large group of researchers working in this field. I judge the manuscript is suitable for publication in Nature Communications.

Reviewer #2 (Remarks to the Author):

The authors have modified their manuscript and in particular they have performed additional experiments and have extended the supplementary material substantially. This supplementary material now makes it much easier for the reader to follow the different steps performed by the authors, starting from the actual experiment throughout the different data analysis steps. I very much appreciate this additional information.

In my previous report, I mostly addressed the issue of the various relevant “scattering rates”. Here the authors also have given additional information, which I find very helpful. But unfortunately, I still have questions concerning this aspect, which I will discuss below. Overall, I find that the experimental results are very interesting and worthy of publication, but in my opinion the authors should reconsider which conclusions can be drawn from the actual data. From my point of view, there are two main observations that are of great interest: a) the temperature dependence of the penetration depth is unusual compared to more conventional superconductors, and its basically quadratic low-temperature limiting behavior suggests axial triplet pairing. b) the temperature dependence of σ_1 below T_c is extremely unusual and suggests a very unconventional state, such as a conducting surface state (but there is no detailed understanding of this behaviour, including the finite value of σ_1 for T approaching zero that exceeds the σ_1 value above T_c). I think that these two statements could be published based on the presented data, whereas the presented additional quantitative analysis I find not sufficiently supported.

I have the following explicit criticisms to the presented manuscript:

- 1) I still do not understand the arguments concerning the scattering rates. The authors have now explicitly labelled different scattering rates, which simplifies the discussion. But I still have the following questions/concerns:
- 2) What is the relation between the “bulk impurity scattering rate Γ_{imp} ” and the normal-state “scattering rate $1/\tau$ ” (section III. of supplementary, e.g. Eq. (S13))? Based on the presented discussion, I would expect that these two are basically the same quantity, maybe with an additional temperature dependence to be taken into account (the temperature dependence visible in Fig. S4), but judging from Fig. S4, the flattening curves for R_s and X_s just above T_c , I would expect that Γ_{imp} in the superconducting state should be on the same order as the normal-state $1/\tau$. With the latter being on the order of 1 THz (p. 11 of supplementary), which is substantially higher than the superconducting gap, one would be in the dirty limit of superconductivity. But this contradicts

the discussion around Eq. (1) of the main paper, which explicitly describes that there is no large bulk impurity scattering rate. (And the fit result " $\Gamma_{\text{imp}} = 0.001 \pm 0.001 \text{ kB T}_c$ " suggests Γ_{imp} around 30MHz, orders of magnitude lower than the $1/\tau$ around 1 THz above T_c .) Please explicitly describe how " $1/\tau$ " (above T_c) and " Γ_{imp} " (below T_c) are related.

3) The fit of Γ_{imp} based on Eq.(1): As far as I see, the temperature dependence of Eq.(1) can be written as $\rho_s(T) = 1 - f(\Gamma_{\text{imp}}, \Delta_0(0)) * T^2$, with a function $f(\Gamma_{\text{imp}}, \Delta_0(0))$ that does not depend on temperature. How can one deduce Γ_{imp} and $\Delta_0(0)$ as independent quantities from such a fit, which contains these two quantities only in the combined prefactor $f(\Gamma_{\text{imp}}, \Delta_0(0))$? To me, such a fit appears meaningless. Please explain this fit procedure.

4) The fit of Γ_{imp} based on Eq.(1): this equation only is valid for $\Gamma/\Delta_0 < 2/\pi$, i.e. being in the clean limit/having a small scattering rate is assumed when applying the formula. Using this formula for fitting thus cannot be used to rule out substantially larger Γ like the 1 THz from the data above T_c .

Smaller comments:

5) Please explicitly define the meaning of " Δ_0 ".

6) From the new details to the experiment, I understand that the "N-grease" was still on the sample during the microwave experiment. Please state this explicitly, as it might cause some obvious questioning: what are the microwave properties of N-grease at mK temperatures? Does this affect your measurements? Are there reference measurements for this? I guess that the answer for the last two questions is "no", but please state it explicitly. Also give explicit information, what "N-grease" actually is.

7) Fig. S5: what does "(rad)" mean as unit for " $\omega\tau$ "? I would expect " $\omega\tau$ " to be unitless.

8) Supplementary, line 268: "However this bulk impurity scattering rate much larger than the critical temperature will suppress the superconducting state even at zero temperature." Please give a reference.

9) In the response letter, page 8, I find the claim " $1/\tau_{\text{tot}} \approx 1/\tau_{\text{surf}} + \Gamma_{\text{imp}}$ " not convincing, because the surface ($1/\tau_{\text{surf}}$) and bulk (Γ_{imp}) scattering rates act on different electronic subsystems and thus contribute to $1/\tau_{\text{tot}}$ in a weighted fashion, and to me it is very unclear how this weighting should be.

10) Discussion of the normal fraction below T_c , see e.g. question (3) of Referee #3 on page 15 of response letter: I agree with Referee #3 that this claim is one of the key statements of the paper (see my introductory comments above). Even if one does not consider the presented technical doubts on the analysis, then I still wonder: how can one explain that σ_1 monotonously increases upon cooling below T_c , down to zero temperature? Could you describe what one would expect if this really were the topological surface state? (Even if this would be speculative?)

Reviewer #3 (Remarks to the Author):

In this revision, the authors addressed some of points raised in the previous referees' report.

Particularly, the critical examination of the possibility of even-parity pairings and the discussion on the residual normal component based on the two-fluid model, which are the main concerns of my previous report, are added in the supplementary material. The former supports the realization of an odd-parity pairing state, excluding the existence of additional line nodes characterizing even parity pairing states. The latter also resolved the question raised in my previous report partly.

The surface-sensitive microwave measurement may probe surface quasiparticle states efficiently. One remarkable feature of the data is that the real part of the conductivity at low temperature is more than two times larger than the normal conductivity at $T=T_c$, as shown in FIG.2b.

The authors attribute this normal component to chiral edge states of the Weyl superconducting state. This result is quite intriguing, and deserve publication in some scientific journals.

However, I cannot yet recommend the manuscript in the current form for publication in Nature Communications because of the following reasons.

It is also possible that the normal component at low temperatures may arise from the pair breaking effect at rough surfaces, and not be an evidence of chiral surface states.

In this case, the strong increase of the conductivity at low temperatures can be understood as a result of the decrease of inelastic scatterings due to electron-electron interaction.

It is desirable to give a discussion about how to exclude this possibility.

Furthermore, according to a symmetry argument, for the realization of non-unitary pairing states in this orthorhombic system, the double transition is necessary; i.e. the direct second-order transition from a paramagnetic normal state to a non-unitary pairing state is not forbidden.

However, there is no experimental signature of such a double transition in this system.

On the contrary, the result shown in FIG.2b indicates that the normal component attributed to chiral surface states appears even just below T_c .

There should be some discussion about how to resolve this point.

I would like to recommend the manuscript for publication provided that the above points are properly addressed.

Response to Referee report

We thank the reviewers for a very careful evaluation of our manuscript. In response to the questions brought up we have designed a new sample preparation method, created a new complementary experiment setup, performed new analysis, and also discuss the consistency of our experimental results with the results from other new experiments on this material. The resulting changes make the arguments presented here considerably stronger. The changes to the manuscript are highlighted in yellow (removed parts are stroked in red) and their locations in the revised manuscript are called out in this response.

Note: Referee's original comments and our response are given in black (Calibri) and green (Times new roman) text, respectively. The page and line number information of the revised content is provided, based on the highlighted manuscript enclosed for review.

Response to the first referee

I have carefully read the revised version of this manuscript, which was previously submitted to [Redacted]. In their Reply the authors have taken away most of the concerns raised in my report. They have also made a number of clarifications in the extensive S M file. Overall, the authors have improved the manuscript. I still find it difficult to digest that other possible causes for the normal contribution are ruled out without much certainty. Especially, the authors rely on Metz et al. to explain the low-temperature contribution in the specific heat as being not relevant in the current experiment. Nonetheless, UTe₂ is certainly an unconventional superconductor. The current paper helps to unravel its nodal gap structure and is of interest to the large group of researchers working in this field. I judge the manuscript is suitable for publication in Nature Communications.

We appreciate the referee's favorable remarks. We would also like to see in the future that our community reveals further understanding on the divergent low-temperature term of the specific heat with more detailed thermal and magnetic studies on this system.

Note that our reliance on [T. Metz, et al., Phys. Rev. B 100, 220504(R) (2019)] to explain the low-temperature contribution in the specific heat as being not relevant is strongly influenced by the fact that their result is based on measurements of the same crystal -- thermal conductivity, showing no residual quasiparticle transport at zero temperature (see plot in response to second Referee), and specific heat showing the low-temperature upturn -- which confirms that the specific heat contribution cannot be itinerant. Also, as discussed at length in response to the referee #3 below, we have improved our consideration of other possible causes of the normal contribution below T_c quantitatively, and increased the certainty in this conclusion.

Response to the second referee

The authors have modified their manuscript and in particular they have performed additional experiments and have extended the supplementary material substantially. This supplementary material now makes it much easier for the reader to follow the different steps performed by the authors, starting from the actual experiment throughout the different data analysis steps. I very much appreciate this additional information.

In my previous report, I mostly addressed the issue of the various relevant “scattering rates”. Here the authors also have given additional information, which I find very helpful. But unfortunately, I still have questions concerning this aspect, which I will discuss below. Overall, I find that the experimental results are very interesting and worthy of publication, but in my opinion the authors should reconsider which conclusions can be drawn from the actual data. From my point of view, there are two main observations that are of great interest: a) the temperature dependence of the penetration depth is unusual compared to more conventional superconductors, and its basically quadratic low-temperature limiting behavior suggests axial triplet pairing. b) the temperature dependence of σ_1 below T_c is extremely unusual and suggest a very unconventional state, such as a conducting surface state (but there is no detailed understanding of this behavior, including the finite value of σ_1 for T approaching zero that exceeds the σ_1 value above T_c). I think that these two statements could be published based on the presented data, whereas the presented additional quantitative analysis I find not sufficiently supported.

I have the following explicit criticisms to the presented manuscript:

1) I still do not understand the arguments concerning the scattering rates. The authors have now explicitly labelled different scattering rates, which simplifies the discussion. But I still have the following questions/concerns:

2) What is the relation between the “bulk impurity scattering rate Γ_{imp} ” and the normal-state “scattering rate $1/\tau$ ” (supplementary note 3, Eq. (13))? Based on the presented discussion, I would expect that these two are basically the same quantity, maybe with an additional temperature dependence to be taken into account (the temperature dependence visible in Fig. S4), but judging from Fig. S4, the flattening curves for R_s and X_s just above T_c , I would expect that Γ_{imp} in the superconducting state should be on the same order as the normal-state $1/\tau$. With the latter being on the order of 1 THz (p. 11 of supplementary), which is substantially higher than the superconducting gap, one would be in the dirty limit of superconductivity. But this contradicts the discussion around Eq.

(1) of the main paper, which explicitly describes that there is no large bulk impurity scattering rate. (And the fit result “ $\Gamma_{imp} = 0.001 \pm 0.001 k_B T_c$ ” suggests Γ_{imp} around 30 MHz, orders of magnitude lower than the $1/\tau$ around 1 THz above T_c). Please explicitly describe how “ $1/\tau$ ” (above T_c) and “ Γ_{imp} ” (below T_c) are related.

We appreciate the referee for raising an important question and we investigated this aspect more thoroughly. In this response, we will first discuss evidence for considering UTe2 to be in the clean limit from the residual thermal conductivity and the absence of T_c variation. After that, we will discuss the relation between Γ_{imp} and $\Gamma_n = 1/\tau_n$ above T_c , and how the estimated small value of Γ_{imp} from the superconducting state is compatible with the large $\Gamma_n = 1/\tau_n \sim 1$ THz in the normal state.

First, unlike s-wave superconductors whose pairing is robust against *nonmagnetic* impurities, the pairs in the nodal, unconventional superconductors such as d- and p-wave superconductors can be easily broken in the presence of *nonmagnetic* impurities [Radtke, et al., Phys. Rev. B 48, 653(R) (1993), MacKenzie, et al., Phys. Rev. Lett. 80, 161(1998)]. For example, an axial p-wave pairing state hosts a significant density of states near zero energy even with very small values of $\Gamma_{imp}/\Delta_0 < 1$ in the Born scattering approximation, as shown below [Sigrist and Ueda, Rev. Mod. Phys. 63, 239 (1991)],

[Redacted]

In such a case, this low energy electronic density of states results in finite residual ($T \rightarrow 0$) linear term of the thermal conductivity (κ/T), whose value is a good indicator of the amount of residual normal electrons. However, as seen from the near zero magnetic field κ/T data (blue curve) below [T. Metz, et al., Phys. Rev. B 100, 220504(R) (2019)], $\frac{\kappa}{T} \rightarrow 0$ as $T \rightarrow 0$, following a nearly quadratic T dependence. This implies that normal electronic states are absent at least in the bulk, and the system is in the clean limit with $\frac{\Gamma_{imp}}{\Delta_0} \ll 1$.

[Redacted]

A second piece of evidence of the clean limit and low bulk Γ_{imp} of this system is the small value of coherence length ξ . $H_{c2}(0)$ of UTe₂ is extremely high (6 ~ 34.5 T depending on axis), yielding extremely short $\xi(0) = 3 \sim 7$ nm. In contrast, considering the Fermi velocity obtained from ARPES data [Miao, et al., Phys. Rev. Lett. 124, 076401 (2020)], the mean free path $l_{mfp}(T_c)$ ($< l_{mfp}(0)$) is estimated to be 340 nm, much larger than the coherence length. In fact, the relation $l_{mfp}(0) \gg \xi(0)$ is the definition of the clean limit for the low temperature regime.

As the referee pointed out, now the natural question arising is how to reconcile the above evidence of the clean limit (bulk $\Gamma_{imp} \ll 2\Delta_0(0)$) and the large scattering rate in the normal state $\Gamma_n \sim 1\text{THz} \gg 2\Delta_0$ obtained from our measurement. In order to answer this question, it would be good to survey the behavior of scattering rates of superconductors from various origins. The scattering process in superconductors consists of elastic scattering such as impurity scattering Γ_{imp} , and inelastic scattering such as electron-electron scattering Γ_{ee} (e.g. due to spin fluctuation), electron-phonon scattering Γ_{e-ph} and other possible mechanisms. The experimentally obtained rate Γ_{tot} is expected to be the sum of these various scattering rates $\Gamma_{tot} = \Gamma_{imp} + \Gamma_{ee} + \Gamma_{e-ph} + \Gamma_{others}$.

An important point here is that scattering rate from different mechanisms can follow different temperature dependence across T_c . For example, while Γ_{imp} does not show noticeable difference across the T_c as seen from Ti (430/490 GHz for below/above T_c) [Thiemann, et al., Phys. Rev. B 97, 214516 (2018)], Γ_{ee} from spin fluctuations shows a very sharp drop just below T_c with T^3 dependence. This sharp drop in Γ_{ee} was theoretically predicted [Quinlan, et al., Phys. Rev. B 49, 1470 (1994)] and experimentally observed in spin fluctuation dominating superconductors such as YBCO [Bonn, et al., Phys. Rev. B 47, 11314 (1993)] and CeCoIn₅ [Truncik, et al., Nat. Commun. 4, 2477 (2013)]. As a result, for systems where the inelastic scattering mechanism dominates $\Gamma_{ee} \gg \Gamma_{imp}$ in the normal state, the ratio of the scattering rate at different temperatures ($\Gamma(T_c)/\Gamma(0)$) can reach up to 40 (CeCoIn₅) and 400 (YBCO). In this case, it is

compatible to have a large total scattering rate in the normal state $\Gamma_{tot}(T_c) = \Gamma_{imp} + \Gamma_{ee}(T_c) \approx \Gamma_{ee}(T_c) \gg 2\Delta_0(0)$, and to have a low impurity scattering rate in the superconducting state $\Gamma_{tot}^{bulk}(0) \approx \Gamma_{imp} \ll 2\Delta_0(0)$ at the same time.

UTe₂ has strong magnetic fluctuations in the normal state as shown from the critical scaling behavior of the magnetization and muon spin relaxation study [Ran, et al., Science 365, 684 (2019), Sundar, et al., Phys. Rev. B 100, 140502(R) (2019)]. Therefore, in UTe₂, it is possible that spin fluctuations are the dominant scattering mechanism in the normal state $\Gamma_{ee}(T_c) \gg \Gamma_{imp}$. In this case, as shown in the above paragraph, having a large $\Gamma_{tot}(T_c) \sim 1$ THz in the normal state ($T > T_c$), and being in the clean limit in the superconducting state $\Gamma_{imp} \ll 2\Delta_0(0)$ (~ 130 GHz) can be reconciled due to the sharp drop in Γ_{ee} just below T_c .

The discussion reconciling the clean limit Γ_{imp} in the 0 K limit and the large scattering rate in the normal state $\Gamma_{tot}(T > T_c)$ has been added to [Supplementary note 3. Page 11, line 225].

3) The fit of Γ_{imp} based on Eq.(1): As far as I see, the temperature dependence of Eq.(1) can be written as $\rho_s(T) = 1 - f(\Gamma_{imp}, \Delta_0(0))T^2$, with a function $f(\Gamma_{imp}, \Delta_0(0))$ that does not depend on temperature. How can one deduce Γ_{imp} and $\Delta_0(0)$ as independent quantities from such a fit, which contains these two quantities only in the combined prefactor $f(\Gamma_{imp}, \Delta_0(0))$? To me, such a fit appears meaningless. Please explain this fit procedure.

We appreciate the referee for pointing out this issue. We originally calculated a root-mean-square error (σ_{RMS}) map between the data and theoretical estimate of $\rho_s(T)$ in terms of two-dimensional parameter space Γ_{imp} and $\Delta_0(0)$. The estimated Γ_{imp} and $\Delta_0(0)$ are the values which give a minimum for σ_{RMS} . However, as the referee pointed out, this procedure is meaningless as both Γ_{imp} and $\Delta_0(0)$, which determine the single coefficient f , are temperature independent. Only the value of f is meaningful from the fitting of $\rho_s(T)$.

Instead, from the value of f , one can determine the relation between Γ_{imp} and $\Delta_0(0)$. The coefficient f is a quadratic function of Γ_{imp} and $\Delta_0(0)$. Solving this quadratic equation gives $\Gamma = 2.21\Delta_0(0) \pm \sqrt{0.17 (\text{meV}^2) + 2.47 \Delta_0^2(0)}$. This results in $\Gamma_{imp} \leq 0.06 \Delta_0(0)$ or $\Gamma_{imp} \geq 4.36 \Delta_0(0)$ for a range of $\Delta_0(0) \leq 0.280$ meV (Note that STM study reports $\Delta_0 = 0.250$ meV at $0.18T_c$, thus $\Delta_0 \leq 0.280$ meV is a safe upper bound). The effect of T_c suppression by Γ_{imp} for a nodal superconductor can be calculated by the

formula $\left(\ln \frac{T_{c0}}{T_c} = \Psi\left(\frac{1}{2} + \frac{\hbar\Gamma_{imp}}{4\pi k_B T_c}\right) - \Psi\left(\frac{1}{2}\right)\right)$, where T_{c0} is the T_c for the case of $\Gamma_{imp} = 0$ [Radtke, et al., Phys. Rev. B 48, 653(R) (1993), Mackenzie, et al., Phys. Rev. Lett. 80, 161(1998)]. According to this, if $\Gamma_{imp} \geq 4.36 \Delta_0(0) \gg k_B T_c$ then T_c will be suppressed by more than 80 % ($T_c = 0.19 T_{c0}$). Such a suppression is not observed for the UTe₂ samples used in our study. Hence, we must conclude that $\Gamma_{imp} \leq 0.06 \Delta_0(0) \ll 2\Delta_0(0)$ is the only physically reasonable solution.

This modified discussion for fitting result of $\rho_s(T)$ to Eq. (1) is now reflected in [main text, page 4, line 110, 113].

4) The fit of Γ_{imp} based on Eq. (1): this equation only is valid for $\Gamma_{imp}/\Delta_0(0) < 2/\pi$, i.e. being in the clean limit/having a small scattering rate is assumed when applying the formula. Using this formula for fitting thus cannot be used to rule out substantially larger Gamma like the 1 THz from the data above T_c .

First of all, in the response for comments (2), we provided examples that it is compatible to have $\Gamma_{tot} \approx \Gamma_{ee} > 1$ THz above T_c and $\Gamma_{imp} \ll 2\Delta_0(0)$ for a spin fluctuation driven superconductor, where UTe₂ shows a clear signature of ferromagnetic fluctuations [Sundar, et al., Phys. Rev. B 100, 140502(R) (2019)].

Secondly, in the low energy regime where the fitting for Γ_{imp} was conducted ($\omega < 0.3 T_c$), the calculated energy profile of the density of states $\rho(\omega)$ of the axial pairing state [Rev. Mod. Phys. 63, 239 (1991)] shows clear sublinear energy (ω) dependence for $\Gamma_{imp}/\Delta_0 = 0.6$.

[Redacted]

However, the representative low energy density of states probes, $\frac{\kappa}{T}(T)$ and $\Delta\lambda(T)$ are quadratic rather than sublinear, which can be seen from $\rho(\omega)$ calculation for $\frac{\Gamma_{imp}}{\Delta_0} < 0.3$ at best. Therefore, the fit of Γ_{imp} based on Eq. (1) of the main text, which is valid for $\frac{\Gamma_{imp}}{\Delta_0(0)} < \frac{2}{\pi} \sim 0.64$ is appropriate.

Smaller comments:

5) Please explicitly define the meaning of “ Δ_0 ”.

In the revised manuscript, “ Δ_0 ” is now explicitly defined in the [main text, page 3, line 87].

6) From the new details to the experiment, I understand that the “N-grease” was still on the sample during the microwave experiment. Please state this explicitly, as it might cause some obvious questioning: what are the microwave properties of N-grease at mK temperatures? Does this affect your measurements? Are there reference measurements for this? I guess that the answer for the last two questions is “no”, but please state it explicitly. Also give explicit information, what “N-grease” actually is.

The Apiezon “N-grease” is the product name of a purified hydrocarbon low temperature adhesive which has low vapor pressure and good thermal conductivity at cryogenic temperatures while being electrically insulating so that it does not affect electrical measurements (<https://apiezon.com/products/vacuum-greases/apiezon-n-grease/>). It is commonly used for cryogenic microwave applications. Regarding its effect on our study, we directly tested the $1/Q$ factor (indicator of dissipation) of our disk dielectric resonator with and without a drop of N-grease present (the same amount we apply for attaching a sample) between 50 mK \sim 2 K. We only see a change in $1/Q$ of $\approx 10^{-6}$, which is near the resolution limit of our $1/Q$ measurement. This is marginal compared to the $1/Q$ with the presence of the sample ($10^{-3} \sim 10^{-4}$). Therefore, we concluded the N-grease does not affect our results with the sample. As the referee recommended, this information is now explicitly described in the [Supplementary note 10, page 20, line 424].

7) Supplementary figure 5: what does “(rad)” mean as unit for “ $\omega\tau$ ”? I would expect “ $\omega\tau$ ” to be unitless.

The revised Supplementary figure 5 now displays $\omega\tau$ as unitless.

8) Supplementary note 5, line 295: “However this bulk impurity scattering rate much larger than the critical temperature will suppress the superconducting state even at zero temperature.” Please give a reference.

We appreciate the referee for pointing out a missing reference. We have now added a corresponding reference [Radtke, et al., Phys. Rev. B 48, 653(R) (1993), MacKenzie, et al., Phys. Rev. Lett. 80, 161(1998)] for the above statement. The first reference predicted a huge suppression of T_c/T_{c0} (here, T_{c0} is the T_c when $\Gamma_{imp} = 0$) for the case of d-wave superconductors as seen in the plot below. Note that the circles are for $T_{c0} = 7.2$ K, boxes

are for 31.6 K, and triangles are for 50.2 K. In all cases, T_c is expected to go nearly zero at least for $\Gamma_{imp} \geq 3k_B T_c$.

[Redacted]

The second reference experimentally observed the rapid suppression of T_c from a nodal superconductor Sr_2RuO_4 even by adding small amount impurities. The T_c drops down to zero eventually when Γ_{imp} becomes on the same order of $k_B T_{c0}$.

[Redacted]

These references are now added to the above sentence in the Supplementary note 5.

9) In the response letter, page 8, I find the claim “ $\frac{1}{\tau_{tot}} \sim \frac{1}{\tau_{surf}} + \Gamma_{imp}$ ” not convincing, because the surface ($1/\tau_{surf}$) and bulk (Γ_{imp}) scattering rates act on different electronic subsystems and thus contribute to $1/\tau_{tot}$ in a weighted fashion, and to me it is very unclear how this weighting should be.

We agree with the referee’s comment that the above expression is misleading. As the referee commented, the sum for effective scattering lifetime τ_{tot} should be made with a weighted fashion (i.e. $\frac{1}{\tau_{tot}} = a_1 \Gamma_{surf} + a_2 \Gamma_{bulk}^{imp} + a_3 \Gamma_{bulk}^{other}$), where $a_{1,2,3}$ are the weighting factors. This is explicitly stated now in revised manuscript [Supplementary note 6 Page 15, line 332].

Note that the value of the weighting factors would depend on the microscopic parameters of the system such as the ratio between the thickness of the surface state (representing the surface fraction) and the magnetic penetration depth (including the bulk fraction). Quantitatively rigorous estimation for those factors is not available yet because it requires a sample thickness dependent broadband measurement when samples with thin-film form become available, as mentioned in [Supplementary note 6 Page 16, line 344].

10) Discussion of the normal fraction below T_c , see e.g. question (3) of Referee #3 on page 15 of response letter: I agree with Referee #3 that this claim is one of the key statements of the paper (see my introductory comments above). Even if one does not consider the presented technical doubts on the analysis, then I still wonder: how can one explain that σ_1 monotonously increases upon cooling below T_c , down to zero temperature? Could you describe what one would expect if this really were the topological surface state? (Even if this would be speculative?)

In the chiral surface normal fluid scenario, the surface normal fluid is topologically protected from direct backscattering by the spin polarized chiral energy dispersion, which increases its scattering lifetime (how much enhancement will be in the ac response is still an open theoretical question). The robustness of this protection is proportional to the gap magnitude $|\Delta_0(T)|$. As T decreases, the thermal fluctuations $k_B T$, which poison this protection, decreases while the $|\Delta_0(T)|$ increases. This provides stronger suppression of the back scattering and enhanced lifetime. We believe the monotonic increase in σ_1 as $T \rightarrow 0$ might be qualitatively understood in this manner. We hope our result motivates the theory community to develop a model for the quantitative understanding of electromagnetic response of the topological surface normal fluid. This remark and qualitative explanation are now added in the perspective part of the manuscript. [main text, page 5, line 146]

Response to the third referee

In this revision, the authors addressed some of points raised in the previous referees' report. Particularly, the critical examination of the possibility of even-parity pairings and the discussion on the residual normal component based on the two-fluid model, which are the main concerns of my previous report, are added in the supplementary material. The former supports the realization of an odd-parity pairing state, excluding the existence of additional line nodes characterizing even parity pairing states. The latter also resolved the question raised in my previous report partly. The surface-sensitive microwave

measurement may probe surface quasiparticle states efficiently. One remarkable feature of the data is that the real part of the conductivity at low temperature is more than two times larger than the normal conductivity at $T=T_c$, as shown in FIG.2b. The authors attribute this normal component to chiral edge states of the Weyl superconducting state. This result is quite intriguing, and deserve publication in some scientific journals. However, I cannot yet recommend the manuscript in the current form for publication in Nature Communications because of the following reasons.

1. It is also possible that the normal component at low temperatures may arise from the pair breaking effect at rough surfaces, and not be an evidence of chiral surface states. In this case, the strong increase of the conductivity at low temperatures can be understood as a result of the decrease of inelastic scatterings due to electron-electron interaction. It is desirable to give a discussion about how to exclude this possibility.

We appreciate the referee for raising this very important question, and here we investigated this possibility carefully. First, from the theoretical point of view, it is discussed that a rough surface may suppress order parameter of an anisotropic superconductor through diffusive scattering at the disordered surface [Barash, et al., Phys. Rev. B 55, 15282 (1997)] with thickness d smaller than coherence length $\xi(0)$ [Bakurskiy, et al., Phys. Rev. B 98, 134508 (2018)]. The surface disorder provides a shorter mean free path l_{mfp} than the bulk, resulting in a large, dominant surface impurity scattering rate Γ_{imp}^{surf} . Once Γ_{imp}^{surf} exceeds a certain critical value, the order parameter at such a surface can be suppressed. However, Γ_{imp} does not show noticeable temperature dependence as seen from Ti (430/490 GHz for below/above T_c) [Thiemann, et al., Phys. Rev. B 97, 214516 (2018)]. This implies that if a surface normal fluid is introduced by a dominant Γ_{imp}^{surf} from the disorder of the rough surface, its contribution to σ_1 would not provide the more than factor of 2 enhancement $\frac{\sigma_1(0)}{\sigma_n} > 2$ observed in our data.

Now, let's consider a case of the clean surface where Γ_{imp}^{surf} is very low so that the inelastic electron-electron scattering rate Γ_{ee}^{surf} , which tends to have a clear temperature dependence, plays the dominant role in surface conductivity. Such an alternative scenario for the pair breaking clean surface is possible with specular reflection in a superconductor whose order parameter changes sign for different directions of the momentum before and after the reflection. Theoretical studies expect this effect for d-wave superconductors to create a normal region (Andreev bound state - ABS) at its surface with a thickness $L \sim \xi$ [Löfwander, et al., Super. Sci. Tech. 14, R53 (2001)]. The consequent experimental signatures of this surface Andreev bound states were observed in YBCO by point-contact spectroscopy [Aprili, et al., Phys. Rev. Lett. 83, 4630 (1999)].

However, the microwave conductivity studies of such systems [Bonn, et al., Phys. Rev. B 47, 11314 (1993), Hosseini, et al., Phys. Rev. B 60, 1349 (1999)] do not show the monotonic increase in σ_1 below T_c observed in our study (Fig. 2(b)). Instead, they show a significant decrease in σ_1 as $T \rightarrow 0$ as seen in the below Fig. R1.

Fig. R1. Real part microwave conductivity $\sigma_1(T)$ of a YBCO crystal much below T_c (~ 89 K) [Hosseini, et al., Phys. Rev. B 60, 1349 (1999)].

[Redacted]

This is possibly because the thickness of the normal layer is very thin ($\xi(0) \approx 2$ nm [Varshney, et al., J. Supercon. 9, 629 (1996)]) compared to penetration depth $\lambda(0) \approx 155$ nm [Tallon, et al., Phys. Rev. Lett. 74, 1008 (1995)] where the electromagnetic response is surveyed. In this case, the normal layer contribution to σ_1 can be marginal. Similarly, UTe₂ has $\xi(0) = \sqrt{\frac{\Phi_0}{2\pi H_{c2}}} = 3 \sim 7$ nm [Ran, et al., Science 365, 684 (2019)] and $\lambda(0) \approx 791$ nm, showing even a smaller ratio of $\frac{\xi(0)}{\lambda(0)}$ than YBCO. This fact makes the contribution of the normal layer from Andreev bound state to σ_1 , even if the layer exists, really marginal.

On the other hand, a naïve, but effective estimation of the thickness of the chiral surface state (d_{CSS}) can be given by the uncertainty principle $\Delta E \Delta t \sim \hbar/2$. As $\Delta t = d_{CSS}/v_F$ and $\Delta E = \Delta_0$, one arrives to $d_{CSS} = \hbar v_F/2\Delta_0$. With $v_F \approx 3.9 \times 10^5$ m/s from ARPES study [Miao, et al., Phys. Rev. Lett. 124, 076401 (2020)] and $\Delta_0 \approx 0.265$ meV, d_{CSS} is estimated to be 488 nm, which is comparable to $\lambda(0) \approx 791$ nm. The chiral surface normal state with thickness d_{CSS} comparable to the microwave probing range $\lambda(0)$ could possibly provide a significant contribution to $\sigma_1(T)$.

These discussions about the pair-breaking surface and chiral surface state have been added in [Supplementary Note 11, Page 21, line 438].

One side note is that, recalling the comment on the first round review at [Redacted] (“Can you provide justification for the seemingly too large residual normal fluid fraction $f_n \sim 0.58$ attributed to the surface Majorana normal fluid?”), now we realize that $\frac{d_{css}}{\lambda(0)} = 0.62$ is very similar to the residual normal fluid fraction $f_n(0) = 0.58$ calculated from two-fluid model analysis (Supplementary note 6, Page 15, line 344), although a more complicated model is required to precisely estimate $f_n(0)$ as discussed in the last two paragraphs of Supplementary note 6.

Another side note regarding the monotonic increase in σ_1 is that, in the surface chiral normal fluid scenario, the surface normal fluid is topologically protected from direct backscattering due to the spin polarized chiral energy dispersion, which will increase its scattering lifetime (by how much for the ac response is still an open question for theoretical study). The robustness of this protection is proportional to the gap magnitude $|\Delta_0(T)|$. As T decreases, the thermal fluctuation $k_B T$, which poisons this protection, decreases while the $|\Delta_0(T)|$ increases, providing stronger suppression of the direct back scattering. Although it requires more quantitative theoretical calculation, we propose that the monotonic increase in σ_1 as $T \rightarrow 0$ might be understood in this manner.

2. Furthermore, according to a symmetry argument, for the realization of non-unitary pairing states in this orthorhombic system, the double transition is necessary; i.e. the direct second-order transition from a paramagnetic normal state to a non-unitary pairing state is not forbidden. However, there is no experimental signature of such a double transition in this system. On the contrary, the result shown in Fig.2b indicates that the normal component attributed to chiral surface states appears even just below T_c . There should be some discussion about how to resolve this point.

As the referee’s comment pointed out, a double transition is necessary on the basis of symmetry arguments. Indeed, recent specific heat study of UTe_2 near T_c showed a clear double transition (below figure) [arXiv:2002.02539].

[Redacted]

One may wonder why this double transition is showing up in the specific heat but not in the microwave conductivity or impedance. We have looked for it in our data but not clearly observed it. The reason is a rather practical measurement issue. The specific heat is proportional to the derivative of the free energy. Therefore, it shows a sharp, “discontinuous jump” across T_c as soon as the order parameter “starts to” develop. Therefore, identifying a double transition is relatively easy even when the two transitions are closely packed as in the case of UTe_2 . However, the changes in the microwave impedance and conductivity are determined by the amount of superfluid density which “gradually” develops below T_c . Therefore, it is much harder to distinguish two transitions from the smooth, continuous curve of the conductivity or impedance near T_c .

Indeed, one can see this difference of the two measurement techniques from another example of a superconductor with two order parameters: UPt_3 . As seen from the below plots, its specific heat shows a clear double transition due to discontinuity across two T_c . However, the microwave resistance does not resolve the two transitions due to the continuous evolution across T_c .

[Redacted]

(left) Specific heat / T near T_c of UPt_3 , clearly showing discontinuous jumps. (right) Normalized microwave surface resistance across T_c , which does not resolve two transitions due to its smooth and gradual change.

Since the specific heat study of UTe_2 resolves a double transition, we believe that two order parameters coexist in this material, hence is able to have a non-unitary pairing state.

As the referee recommended, a remark and reference to the two transitions observed in specific heat study have been added in [main text, page 4, line 94] for the readers who would like to see consistency with the group theory analysis.

I would like to recommend the manuscript for publication provided that the above points are properly addressed.

REVIEWERS' COMMENTS

Reviewer #2 (Remarks to the Author):

The authors have very thoroughly responded in their letter to my previous questions, and I very much appreciate their discussion of the various relevant points, which was quite helpful for me. They have also modified the manuscript and the supplementary material in several sections, and I in my opinion this has further improved the manuscript.

Now all presented claims on the unconventional properties of the superconducting phase of UTe₂ are robust, and they will help for the understanding of this material of substantial present interest.

With these changes, I recommend publication in Nature Communications, and I wish the authors positive response from the scientific community and further insights from future investigations of UTe₂.

I just have one further minute optional suggestion: On page 4 of the manuscript, line 115, it is stated that "For a nodal superconductor, the large [...] suppresses T_c more than 80 %. Such suppression was not observed for the sample in this study." I guess that this is supposed to mean that because T_c is not suppressed for the particular sample, Gamma_{imp} has to be the smaller solution of two possibilities. But this suggests that the quantitative T_c suppression for UTe₂ due to defects etc. is already experimentally established, similar to Ref [21] for Sr₂TiO₄. If there is such a study for UTe₂, please give a reference.

Reviewer #3 (Remarks to the Author):

I have read carefully the responses from the authors and the revised manuscript.

In this revision, all the points raised in my previous reports are fully addressed by adding the precise discussions on pair-breaking effects on surfaces of the sample, and the absence of the signature of the double transition in the microwave measurements. I am quite satisfied with the authors' responses.

Now, I would like to recommend the manuscript for publication in Nature communications.

Second Response to Referee report

We thank the reviewers for a very careful evaluation of our manuscript. In response to the question brought up, we have added relevant references.

Response to the second referee

The authors have very thoroughly responded in their letter to my previous questions, and I very much appreciate their discussion of the various relevant points, which was quite helpful for me. They have also modified the manuscript and the supplementary material in several sections, and I in my opinion this has further improved the manuscript.

Now all presented claims on the unconventional properties of the superconducting phase of UTe_2 are robust, and they will help for the understanding of this material of substantial present interest.

With these changes, I recommend publication in Nature Communications, and I wish the authors positive response from the scientific community and further insights from future investigations of UTe_2 .

I just have one further minute optional suggestion: On page 4 of the manuscript, line 115, it is stated that “For a nodal superconductor, the large [...] suppresses T_c more than 80 %. Such suppression was not observed for the sample in this study.” I guess that this is supposed to mean that because T_c is not suppressed for the particular sample, Γ_{imp} to be the smaller solution of two possibilities. But this suggests that the quantitative T_c suppression for UTe_2 due to defects etc. is already experimentally established, similar to Ref. [21] for Sr_2RuO_4 . If there is such a study for UTe_2 , please give a reference.

We appreciate the referee’s favorable remarks. The referee’s constructive comments have helped us to improve the discussion of scattering rate more thoroughly compared to the initial submission. We appreciate the referee’s invaluable effort in reviewing our work.

To address the referee’s last suggestion, a reference [Aoki et al., J. Phys. Soc. Jpn. 88, 043702 (2019)] is now added, which shows the flux-grown UTe_2 samples with gradual suppression of T_c all the way down to < 100 mK as the normal state dc resistivity increases. On the other hand, the samples grown by chemical vapor transport, including the crystal used in our study, shows consistent $T_c \sim 1.6$ K as seen from [Hayes et al., arXiv:2002.02539 (2020), Ran, et al., Science 365, 684 (2019), and others]. These references are now added in the main text, page 5 of the manuscript as below.

“Such suppression was not observed for the samples grown by the chemical vapor transport method [Ran, et al., Science 365, 684 (2019), Hayes et al., arXiv:2002.02539 (2020)]. They show a consistent $T_c \sim 1.6$ K, including the sample in this study. In

contrast, samples prepared by the flux-growth method [Aoki et al., J. Phys. Soc. Jpn. 88, 043702 (2019)] show suppression in T_c all the way down to < 0.1 K, as their normal state dc resistivity increases.”

Response to the third referee

I have read carefully the responses from the authors and the revised manuscript. In this revision, all the points raised in my previous reports are fully addressed by adding the precise discussions on pair-breaking effects on surfaces of the sample, and the absence of the signature of the double transition in the microwave measurements. I am quite satisfied with the authors’ responses. Now, I would like to recommend the manuscript for publication in Nature communications.

We appreciate the referee’s favorable remarks. The referee’s constructive comments guided us to check the other possible scenarios more comprehensively, which helped the conclusion of this work to be more robust. We appreciate the critical comments and precious time spent reviewing our work.